# LEARNING MULTI-MODAL REPRESENTATION ALIGNMENTS FROM NOISY DATA-PAIRS

## ABSTRACT

Contrastive learning (CL) represents one of the most successful paradigms for self-supervised representation learning, which has been applied to SOTA multi-modal learning applications. One overlooked limitation of standard contrastive learning, however, is that it is not designed for robust learning in the presence of noisy data pairs. For example, not all negative samples are truly negative, *e.g.*, within a mini-batch there can be negative samples that are semantically as positive as the positive sample. This is common in most web-sourced multi-modal datasets such as CC3M and YFCC that are frequently used for CL, due to the noisy nature when crawling the datasets. Consequently, the noise in the datasets could significantly impair the power of CL. To remedy this issue, we propose a novel solution by reformulating the standard CL into a probability framework, and introducing learnable random weights to associate with data pairs, so as to allow automatic inference of the degree of noisiness for each data pair. Within our probability framework, posterior inference of the random weights can be done efficiently with Bayesian data augmentation. Consequently, the model can be effectively optimized by a novel learning algorithm based on stochastic expectation maximization. We demonstrate the effectiveness of our approach on several standard multi-modal contrastive learning benchmarks, which significantly outperforms standard contrastive learning.

## 1 INTRODUCTION

Contrastive learning has become increasingly popular in multi-modal representation learning due to its effectiveness in aligning representations from different modalities. In the context of vision-language representation learning, the model aims to learn generic representations from images and texts that could benefit multi-modal downstream applications such as zero-shot image classification and image-text retrieval. Recent advances (Radford et al., 2021; Jia et al., 2021; Li et al., 2021; Zhou et al., 2022; Gao et al., 2023; Guo et al., 2023) have scaled up vision language representation learning by applying contrastive loss to pre-train the model with a substantial volume of web-sourced paired image-text data such as Conceptual Caption (Sharma et al., 2018), YFCC (Thomee et al., 2016), Laion (Schuhmann et al., 2022). While some studies combine the representations of two modalities into a single encoder (Wang et al., 2021a; 2022b;c; 2021b), it is more prevalent to represent the image and text modalities separately using modality-specific encoders similar to the CLIP framework (Mokady et al., 2021; Shen et al., 2021; Jia et al., 2021; Li et al., 2021; Duan et al., 2022; Yang et al., 2022; Shukor et al., 2022). After pre-training, the model can produce general representations of both image and text inputs, demonstrating exceptional performance in subsequent tasks. Recent advances show that these high-quality representations can be adapted to text-guided generation of natural images (Ramesh et al., 2021; Crowson et al., 2022; Xu et al., 2023; Ruiz et al., 2023; Liu et al., 2023), videos (Kwon et al., 2022; Lin et al., 2022; Rasheed et al., 2023), 3D shape (Sanghi et al., 2023; Wang et al., 2022a; Sanghi et al., 2022), point clouds (Zhu et al., 2022), and semantic segmentation (Park et al., 2022; Zhou et al., 2023; Liang et al., 2023), etc.

In multi-modal representation learning, standard contrastive loss seeks to maximize the similarity between corresponding image-text pairs (termed "positive pairs") while distinguishing them from all the non-matching image-text pairs (termed "negative pairs"). Such an objective aligns the true image-text pairs together to build meaningful representations. Although contrastive loss has proven effective in empirical applications for multi-modal representation learning, there remain two open

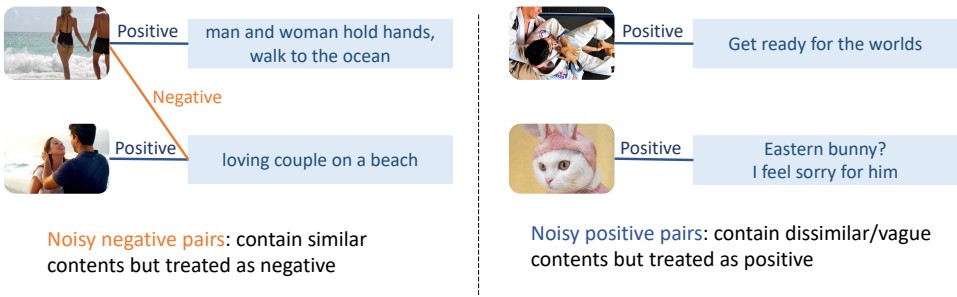

Figure 1: Examples from CC3M (Sharma et al., 2018) dataset that contain noisy pairs.

questions that have been largely ignored in previous works. First, are the ground truth labels of "positive" and "negative" from the web-sourced dataset truly reliable? Most common web-sourced datasets consider images and their corresponding descriptions as the **only** true positive pairs. Yet in those datasets there can be multiple image-text pairs containing similar contents while being labeled as negative pairs. In other words, web-sourced datasets, due to their large volume and automated collection processes without human labeling, naturally contain substantial noisy pairs. For example, in Figure 1, the first image is considered as a true positive to the text "*man and woman hold hands, walk to the beach*". Both the other texts in the same batch would be considered as negative samples that should be pushed away from the representation of the image. However, the second text "*loving couple on a beach*" can also be considered semantically positive to reflect the content of the first image, while being labeled as "negative" during training. In addition, there also can be other positive pairs in the dataset that contain dissimilar or vague descriptions such as the right example in Figure 1. Such noisy data pairs could potentially lead to mixed training signals and loss of accuracy in performance.

The second open question is whether contrastive learning can handle such noisy pairs. The design of conventional contrastive learning amplifies the importance of true positive pairs within every mini-batch and pushes away all the negative pairs equally. Thus it could be susceptible to inconsistent training signals. For instance, in Figure 1, although the second text contains more similar content to the image, it is treated equally "negative" as other texts in the same batch. Without the flexibility to adjust the importance of each data pair, contrastive learning could overfit into the noisy data pairs within the web-source dataset, leading to sub-optimal solutions.

To address these limitations, we propose a fundamental approach to incorporate stochastic weighting into contrastive learning. Specifically, we augment the contrastive loss by assigning a probability weight to each data pair to allow automatic inference on the degree of nosiness level of the pair. By doing so, we can imbue the system with a degree of flexibility, allowing it to better discern and adapt to the inherent uncertainties in the data. This ensures that data pairs are treated more accurately based on their likelihood of being genuine positive or negative pairs, rather than relying on potentially erratic batch-specific determinations. For efficient learning and inference, we first reformulate the problem into a probability framework with Bayesian data augmentation. The formulation allows us to efficiently infer the weight of each data pair in contrastive learning, such that the learned representation is robust towards noisy training data. Finally, we develop a stochastic expectation maximization algorithm to incorporate the inferred random weights for efficient learning of model parameters. To summarize, our paper has the following major contributions:

- For the first time, we identify the inherent noise problem for some most commonly-used datasets for contrastive learning, and formulate the problem as contrastive learning with noisy data pairs.

- We propose a principled method to solve the problem by reformulating it into a probability framework with Bayesian data augmentation techniques. Based on the reformulation, a novel stochastic expectation maximization algorithm is developed to effectively learn the robust model while simultaneously inferring the stochastic data-pair weights.

- With extensive and large-scale experiments, we demonstrate improved performance on several public benchmarks for multi-modal contrastive learning.

## 2 METHOD

We start by describing the basic setup and notation in contrastive learning, where a backbone network, parameterized by $\boldsymbol{\theta}$, is used to generate generalized representations, written as $\mathbf{z} = \mathsf{enc}(\mathbf{x}; \boldsymbol{\theta})$ for input data $\mathbf{x}$. The multi-modal data is represented in terms of positive and negative data pairs. Specifically, given a multi-modal dataset $\mathcal{D} \triangleq \{(\mathbf{x}_i^1, \mathbf{x}_i^2)\}$ where the superscript indexes different modalities and subscript indexes data samples, each $(\mathbf{x}_i^1, \mathbf{x}_i^2)$ represents a positive pair and each $(\mathbf{x}_i^1, \mathbf{x}_j^2)$ with $i \neq j$ represents a negative pair. Denote $s_{i+} \triangleq \mathsf{sim}(\mathsf{enc}(\mathbf{x}_i^1; \boldsymbol{\theta}), \mathsf{enc}(\mathbf{x}_i^2; \boldsymbol{\theta}))$ as the similarity score between the positive pair $(\mathbf{x}_i^1, \mathbf{x}_i^2)$ after the encoder; and $s_{ik-} \triangleq \mathsf{sim}(\mathsf{enc}(\mathbf{x}_i^{m_1}; \boldsymbol{\theta}), \mathsf{enc}(\mathbf{x}_k^{m_2}; \boldsymbol{\theta}))$ as the similarity score between the negative pair $(\mathbf{x}_i^{m_1}, \mathbf{x}_k^{m_2})$, where $m_1, m_2 \in \{1, 2\}$ and $\mathsf{sim}(\cdot, \cdot)$ denotes a similarity metric (positive value). We adopt the exponential cosine similarity used in most contrastive learning methods in this paper, i.e., $\mathsf{sim}(\mathbf{x}_1, \mathbf{x}_2) \triangleq e^{\mathbf{x}_1^T \mathbf{x}_2}$. Note the similarity scores depend on the model parameter $\boldsymbol{\theta}$, but we omit it in our development for notation simplicity.

### 2.1 PROBABILITY WEIGHTED CONTRASTIVE LEARNING

As discussed in the Introduction, contrastive learning is designed specifically for the ideal case of clean pair data. Specifically, consider the standard setup with one positive pair and $K$ negative pairs for each data sample. The contrastive loss is defined as:

$$\mathcal{L}_{\text{con}}(\mathcal{D}; \boldsymbol{\theta}) = -\frac{1}{|\mathcal{D}|} \sum_{\mathbf{x}_i \in \mathcal{D}} \log(\mathcal{L}_{\mathbf{x}_i}), \text{ with } \mathcal{L}_{\mathbf{x}_i} \triangleq \frac{s_{i+}}{s_{i+} + \sum_{k=1}^{K} s_{ik-}} \ .$$

However, real data usually come with noisy pairs, rendering directly applying contrastive learning problematic. In the following, we describe our fundamental method to deal with such a noisy pair data setting for contrastive representation learning. Our basic idea is intuitive, which is to generalize the standard contrastive loss by adding learnable stochastic weights for all the data pairs. Specifically, we introduce local learnable weights $\{w_i^+, w_{ik}^-\}$ associated with the data pairs, and define the following noise-robust weighted contrastive loss:

$$\mathcal{L}_{\text{con}}^r(\mathcal{D}; \boldsymbol{\theta}) = -\frac{1}{|\mathcal{D}|} \sum_{\mathbf{x}_i \in \mathcal{D}} \log(\mathcal{L}_{\mathbf{x}_i}^r), \text{ with } \mathcal{L}_{\mathbf{x}_i}^r \triangleq \frac{w_i^+ s_{i+}}{w_i^+ s_{ij+} + \sum_{k=1}^{K} w_{ik}^- s_{ik-}} \ , \tag{1}$$

where $\{w_i^+\}$ represents weights for positive pairs, and $\{w_{ik}^-\}$ for negative pairs. Note when considering all weights to be equal to one, the loss reduces to the standard contrastive loss.

One challenge with such a loss, however, is that these auxiliary random weights are local random variables that grow quadratically w.r.t. the training data size (including augmented data), which is essentially infinite and thus infeasible to be stored in the setting of continuous data augmentation. To overcome the challenge, inspired by the recent probability reformulation of contrastive learning (Chen et al., 2022), we propose a scalable Bayesian-learning mechanism to efficiently sample the local weights in each iteration, which are then integrated into the contrastive loss to optimize the global model parameter.

Specifically, we reformulate the problem from a Bayesian inference perspective, where we assign appropriate priors for the weights. We can consider either Bernoulli priors to model weights as binary random variables, or Gamma priors to model them as positive values. For modeling convenience, we consider Gamma priors, i.e.,

$$w_i^+ \sim \mathsf{Gamma}(a_+, b_+), \ \ w_{ik}^- \sim \mathsf{Gamma}(a_-, b_-) \ ,$$

where $a_+$ and $a_-$ are the shape parameters, and $b_+$ and $b_-$ are the rate parameters. This gives a joint posterior distribution over the global model parameter and local random weight variables $w_i^+$ and $w_{ik}^-$, as

$$p(\{w_i^+\}, \{w_{ik}^-\}, \boldsymbol{\theta}; \mathcal{D}) \propto \prod_{\mathbf{x}_i \in \mathcal{D}} \frac{w_i^+ s_{i+}}{w_i^+ s_{ij+} + \sum_{k=1}^{K} w_{ik}^- s_{ik-}} p(\{w_i^+\}) p(\{w_{ik}^-\}) p(\boldsymbol{\theta}) \ .$$

This probability weighting mechanism can be seen as a measure of confidence in the pairing, offering a more flexible and adaptive learning process. It can accommodate the variations and possible inconsistencies in the data, allowing the model to better adapt to real-world complexities.

Another challenge, however, is that directly performing Bayesian inference on such a posterior distribution is infeasible, due to the non-conjugacy between the priors and likelihood. Fortunately, we can borrow ideas from Chen et al. (2022) to introduce an augmented random variable $u_i$ to associate to each data point, giving us an augmented joint posterior distribution equivalent to $p(\{w_i^+\}, \{w_{ik}^-\}, \boldsymbol{\theta}|\mathcal{D})^*$, as

$$p(\boldsymbol{\theta}, \mathbf{u}, \mathbf{w} \,|\mathcal{D}) \propto \prod_{i:\mathbf{x}_i \in \mathcal{D}} w_i^+ s_{i+} e^{-\mathbf{u}_i w_i^+ s_{i+}} \prod_k e^{-u_i w_{ik}^- s_{ik-}} p(\{w_i^+\})p(\{w_{ik}^-\})p(\boldsymbol{\theta}) \,, \qquad (2)$$

where $\mathbf{u} \triangleq \{u_1, u_2, \cdots, u_{|\mathcal{D}|}\}$ and $\mathbf{w} \triangleq \{w_i^+\} \cup \{w_{ik}^-\}$. Consequently, we can perform learning and inference based on the augmented posterior of $p(\boldsymbol{\theta}, \mathbf{u}, \mathbf{w} \,|\mathcal{D})$. In the following, we propose an efficient algorithm based on stochastic expectation maximization (stochastic EM) to alternatively infer the local random variables and optimize the global model parameter.

## 2.2 Efficient Inference and Learning with Stochastic Expectation Maximization (Stochastic EM)

Based on the idea in Chen et al. (2022), we propose a stochastic EM algorithm for efficient inference and learning of our model. Stochastic EM is a stochastic variant of the popular EM algorithm, which alternatively infers local random variables and optimizes global model parameters for a latent variable model (Allassonnière & Chevallier, 2021; Chen et al., 2018; Delyon et al., 1999). It consists of three steps: *simulation, stochastic approximation, and maximization*. In our setting, *simulation* corresponds to sampling local random variables for a batch of data, *e.g.*, $\mathbf{u}$ and $\mathbf{w}$; *stochastic approximation* then uses the sampled auxiliary random variables to update a stochastic objective $Q(\boldsymbol{\theta})$ at each iteration $t$ as: $Q_{t+1}(\boldsymbol{\theta}) = Q_t(\boldsymbol{\theta}) + \lambda_t(\log p(\boldsymbol{\theta}, \mathbf{u}, \mathbf{w} \,|\mathcal{D}) - Q_t(\boldsymbol{\theta}))$, where $\{\lambda_t\}$ is a sequence of decreasing weights; Finally, in *maximization*, we optimize the model parameter $\boldsymbol{\theta}$ by maximizing the stochastic objective $Q_{t+1}(\boldsymbol{\theta})$. We describe more details below:

**Simulation**   Given the joint posterior distribution in equation 2 and the current batch of data, one can easily sample the local random variables $\mathbf{u}$ and $\mathbf{w}$, which simply follow Gamma distributions of the following forms:

$$u_i|\{w_i^+, w_{ik}^-, \boldsymbol{\theta}\} \sim \mathsf{Gamma}(a_u, b_u + w_i^+ s_{i+} + \sum_k w_{ik}^- s_{ik-}), \;\; \forall i, \text{ and} \qquad (3)$$

$$w_i^+|\{\mathbf{u}, \boldsymbol{\theta}\} \sim \mathsf{Gamma}(1 + a_+, u_i s_{i+} + b_+), \text{ and } w_{ik}^-|\{\mathbf{u}, \boldsymbol{\theta}\} \sim \mathsf{Gamma}(a_-, u_i s_{ik-} + b_-), \forall i, k$$

These sampled random variables for the current batch of data will then be used in the stochastic approximation step described below. Optionally, to make the algorithm more stable, we propose to update $u_i$'s with moving averages after sampling, *e.g.*, we maintain $\{u_i\}$ in the memory and update them as: $u_i \leftarrow \alpha u_i + (1 - \alpha)\tilde{u}_i$, where $\tilde{u}_i \sim \mathsf{Gamma}(a_u, b_u + w_i^+ s_{i+} + \sum_k w_{ik}^- s_{ik-})$ and $\alpha \in [0, 1]$ is a hyper-parameter to balance old and new values. This strategy only requires limited storage overhead as we only need extra memory proportional to the training data size, which is considered negligible compared to other parameters.

**Stochastic approximation**   We then proceed to calculate the stochastic approximation based on the simulated local random variables above. For notation simplicity, we define $Q_0(\boldsymbol{\theta}) = 0$. Then we can reformulate $Q_{t+1}(\boldsymbol{\theta})$ by decomposing the recursion, resulting in

$$Q_{t+1}(\boldsymbol{\theta}) = \sum_{\tau=0}^t \tilde{\lambda}_\tau \log p(\boldsymbol{\theta}, \mathbf{u}_\tau, \mathbf{w}_\tau \,|\mathcal{D}_\tau), \text{ where } \tilde{\lambda}_\tau \triangleq \lambda_\tau \prod_{t'=\tau+1}^t (1 - \lambda_{t'}) \,, \qquad (4)$$

where $\tau$ indexes the minibatch and the corresponding local random variables at the current time $\tau$.

---

*In the sense that marginalizing over the augmented random variables $\{w_i^+\}$ and $\{w_{ik}^-\}$ in $p(\theta, \mathbf{U}, \{w_i^+\}, \{w_{ik}^-\}|\mathcal{D})$ gives back to the original $p(\{w_i^+\}, \{w_{ik}^-\}, \boldsymbol{\theta}; \mathcal{D})$. Thus, learning and inferences on the two forms are equivalent.

---

**Algorithm 1** Noise-Robust Contrastive Learning with Stochastic EM

---

1: Initialize $\boldsymbol{\theta}$; set $t = 1$
2: **for** $\mathbf{x}_1, \mathbf{x}_2$ in loader **do**                    $\triangleright$ load a minibatch $(\mathbf{x}_1, \mathbf{x}_2)$ with $B$ samples
3:     Calculate positive/negative similarity scores $\{s_{i+}\}$ and $\{s_{ik-}\}$
4:     Initialize all the weights $\{w_i^+\}$ and $\{w_{ik}^-\}$ to be one
5:     **for** $k = 1 \cdots$ iter [2 in practice] **do**
6:         Sample $\mathbf{u}$ according to equation 3
7:         Sample $\mathbf{w}$ according to equation 3
8:     **end for**
9:     Calculate the weighted contrastive loss in equation 1 with the sampled $\mathbf{w}$ on the current batch of data
10:     Update the model parameter by stochastic gradient descent with the calculated weighted contrastive loss
11:     $t = t + 1$
12: **end for**

---

**Maximization**   The stochastic approximation objective in equation 4 provides a convenient form for stochastic optimization over time, similar to online optimization (Bent & Van Hentenryck, 2005). Specifically, at each time $t$, we can initialize the parameter $\boldsymbol{\theta}$ from the last step, and update it by stochastic gradient descent calculated from the current batch of data. To reduce variance, we propose to optimize a marginal version of $p(\boldsymbol{\theta}, \mathbf{u}_\tau, \mathbf{w}_\tau \,|\mathcal{D}_\tau)$ by integrating out $\mathbf{u}_\tau$, which essentially reduces to our original weighted contrastive loss in equation 1.

With the above steps, it is ready to optimize the model by stochastic EM. The details are provided in Algorithm 1.

## 3   RELATED WORKS

**Vision-Language Representation Learning:**  Recent advances in vision-language representation learning can be broadly classified based on the manner in which information from two modalities is utilized for joint learning. The first category leverages unified models (Wang et al., 2021a; 2022b;c; 2021b) to process both images and texts. Typically, these inputs are tokenized into sequences (Peng et al., 2022; Bao et al., 2022). The latter methods deploy separate encoders (Radford et al., 2021; Mokady et al., 2021; Shen et al., 2021; Li et al., 2021; Duan et al., 2022; Yang et al., 2022; Shukor et al., 2022; Kwon et al., 2022; Jia et al., 2021) for images and texts. To align the different modalities, they utilize the contrastive loss (Oord et al., 2018; He et al., 2020; Chen et al., 2020). It's noteworthy that these techniques have been demonstrated to achieve state-of-the-art (SOTA) results on multiple downstream tasks. How to obtain robust and representational embeddings from CL is vital to benefit downstream tasks. Specifically, we focus on how to cope with noisy positive-negative pairs for CL.

**Noisy Pairs in Contrastive Learning:**  While most works directly utilize large scale dataset for contrastive learning, some argue the noisy dataset issue. Noisy contrastive learning is an advanced technique that addresses the challenges of standard contrastive learning when faced with inconsistencies or "noise" within paired data. Traditional contrastive methods often struggle with mislabeled or ambiguous pairs, leading to decreased accuracy and efficiency. Noisy contrastive learning, on the other hand, incorporates mechanisms, often probabilistic in nature, to accommodate these uncertainties. By assigning confidence or probability weights to each pair, this approach allows for more adaptive and flexible learning. Rather than being limited by the binary classification of pairs, it embraces the inherent complexities and variations in real-world data, enhancing the model's robustness and performance. NLIP (Huang et al., 2023) enforces the pairs with larger noise probability to have fewer similarities in embedding space to improve the model training. Han et al. (2022) apply noise estimation component to adjust the consistency between different modalities for the action recognition task. RINCE (Hoffmann et al., 2022) uses a ranked ordering of positive samples to improve InfoNCE loss. Another recent work (Chen et al., 2022) studies the gradient bias issue in contrastive learning and proposes a stochastic approach to levitate it. To combat the gradient bias, the authors introduce a Bayesian data augmentation approach. This new method transforms the contrastive loss into a decomposable form. Consequently, conventional stochastic optimization can be applied with-

out inducing gradient bias. Our approach uses a stochastic approach from a different perspective to address the noisy data issue. To combat this challenge, we are introducing a probability extension. This innovative approach assigns a probability weight to each pair, whether positive or negative. By doing so, the model is no longer rigidly committed to a binary classification of the pairs but can now take into consideration the uncertainties or noise present in the data. This not only provides more nuanced information to the model but also enhances its robustness.

**Stochastic Expectation Maximization** Stochastic EM (Nielsen, 2000) stands as a pivotal algorithm in machine learning and probabilistic modeling. Building upon the foundations of the classical Expectation-Maximization (EM) algorithm (Lin, 2011), Stochastic EM offers an efficient solution for parameter estimation in situations involving vast datasets or latent variables, *e.g.*, to maximize the log-likelihood of $p(\mathbf{z}, \mathcal{D}|\boldsymbol{\theta})$, where $\mathcal{D}$ is the dataset, $\mathbf{z}$ is the local random variable and $\boldsymbol{\theta}$ is the global model parameter. By leveraging the power of mini-batch sampling, Stochastic EM strikes a balance between computational scalability and estimation accuracy. It has found widespread utility in various domains, including clustering (Allassonnière & Chevallier, 2021), topic modeling (Zaheer et al., 2016), and latent variable modeling (Zhang & Chen, 2020), making it an indispensable tool to cope with complex probabilistic models and extensive data and a natural fit to our problem.

## 4 EXPERIMENTS

We conduct experiments focused on image-text contrastive learning using CLIP-based models, wherein two distinct encoders are trained to align features between image and text modalities. We then evaluate on standard benchmarks including zero-shot, distribution shift, and linear probing tasks. We also provide ablation study and analysis on the sampling hyper-parameters and sampled weights.

### 4.1 EXPERIMENTS SETUP

For encoders, our CLIP model adopts ResNet-50 (He et al., 2016) as the image encoder and BERT (Devlin et al., 2018) as the text encoder. We adopt the official code from OpenCLIP (Ilharco et al., 2021) and DeCL (Chen et al., 2022) to reproduce the baselines and our methods. Our reproduced CLIP results are consistent with the recent works (Mu et al., 2021; Gao et al., 2021; Duan et al., 2022; Jiang et al., 2023), although their results are slightly lower than reported in the original CLIP paper. One possible reason is that we use fewer GPUs, thus leading to a smaller effective batch size. It is important to highlight that all the methods adopt the same OpenCLIP codebase and identical hyper-parameter configurations, thus ensuring a fair comparison.

**Pre-training:** We follow the standard practice and pre-train the model with the CC3M (Sharma et al., 2018) dataset with 3M unique images and 4M image-text pairs.

**Evaluation:** For zero-shot image classification evaluation, we take the pre-trained image encoder to obtain image representation, as well as the pre-trained text encoder and prompts to construct class descriptions to obtain class representations. We evaluate on ImageNet for embedding quality and its distribution shifted benchmarks to evaluate the robustness of our methods. We further evaluate linear probing performance, where the encoders are fixed and one linear layer is trained with additional supervision to evaluate the quality of the learned representations

**Implementation Details:** We follow the same code base and hyper-parameters setting as OpenCLIP except for the number of GPUs. We train the model from scratch on 8 NVIDIA V100 GPUs for 32 epochs. Our batch size is set to 128 per GPU and the feature dimension is 1024. We use an initial learning rate of $5e^{-4}$. We warm up the learning rate for 10000 iterations and follow the cosine decay scheduling. AdamW (Loshchilov & Hutter, 2019) optimizer is used along with a weight decay of 0.2. To further demonstrate the effectiveness of our approach for noisy datasets, we add a random noise of 10% into the training data by randomly selecting 10% of data pairs within every batch and re-sample the positive labels such that 10% of the training data has incorrect positive pairs. For all the baselines we use the same codebase to train from scratch with fixed random seed and the same hyper-parameters for fair comparisons. After pre-training, we evaluate the model trained on the last epoch for all baselines and our approach.

Table 1: Zero-Shot Transfer Learning Classifiction Accuray (%) on ImageNet1K.

| Method | Top1 Accuracy ↑ | Top5 Accuracy ↑ |
|---|---|---|
| CLIP | 17.71 | 35.87 |
| DeCL | 17.55 | 36.46 |
| OURS | **20.96** | **38.24** |

Table 2: Zero-Shot Natural Distribution Shift Classifiction Accuray (%).

| Method | ImageNetV2 | | ImageNetSketch | | ImageNet-A | | ImageNet-R | |
|---|---|---|---|---|---|---|---|---|
| | Top1 ↑ | Top5 ↑ | Top1 ↑ | Top5 ↑ | Top1 ↑ | Top5 ↑ | Top1 ↑ | Top5 ↑ |
| CLIP | 16.44 | **34.15** | 10.23 | 24.21 | **5.05** | **17.71** | 24.75 | 46.30 |
| DeCL | 15.58 | 33.11 | 10.1 | 22.57 | 3.94 | 15.66 | 22.68 | 44.26 |
| OURS | **17.63** | 33.25 | **12.36** | **25.76** | 4.21 | 14.76 | **25.85** | **46.42** |

## 4.2 ZERO-SHOT TRANSFER LEARNING EVALUATION

We conduct zero-shot transfer on standard image classification tasks using the ImageNet1K dataset (Russakovsky et al., 2015). We employ the standard evaluation strategy of prompt engineering. For each dataset, we construct text prompts using the name of the class with some templates, for example, "a photo of the [class name]" and "a sketch of the [class name]". We obtain the normalized class text embedding for each class with multiple standard prompts. We obtain the image embeddings from the pre-trained encoder. During evaluation, the class whose text embedding has the highest similarity score to the image embedding is used as the prediction of the label. Consistent with previous works, we report Top-K classification accuracy with $K = 1, 5$.

We show in Table 1 the zero-shot transfer learning performance, we include other baselines for reference while we mainly focus on comparing with CLIP and DeCL. DeCL improves the clip baseline performance by 1% on Top5 accuracy by solving the gradient bias issue, while our approach can improve over CLIP by 3% on both Top1 and Top5 accuracy with stochastic training pairs re-weighting. Note that both DeCL and our method do not require additional computing except for the sampling processes compared to the original CLIP baseline, which is negligible relative to the total training cost.

## 4.3 NATURAL DISTRIBUTION SHIFT EVALUATION

We also assess variations of the ImageNet1K datasets with featuring shifted distributions (Recht et al., 2019; Wang et al., 2019; Hendrycks et al., 2021b;a). These datasets incorporate sketches, cartoons, and adversarially generated images. They are usually considered as domain-shifted versions of ImageNet and are frequently utilized to evaluate the generalizability and robustness of models, as they usually contain harder or less common data samples. We perform the zero-shot evaluation using the same processes mentioned in the previous section and report classification accuracy on Top-1 and Top-5.

We show in Table 2 the zero-shot transfer learning performance on the Natural Distribution Shift benchmark. We can see that DeCL performs the worst on all four benchmarks, while CLIP baseline demonstrates the best performance on ImageNet-A. CLIP also features decent performance on Top5 accuracy for ImageNetV2. Our method improves the clip baseline performance by 1-2% on Top1 accuracy for three out of four benchmarks (ImageNet-V2, ImageNetSketch, and ImageNet-R), and by around 1% on two out of four benchmarks (ImageNetSketch and ImageNet-R). This indicates that by using our approach to weight training pairs with stochastic approximation we are able to improve the robustness and generalizability of the learned embeddings. Interestingly, our method under-performs CLIP on ImageNet-A, a dataset with adversarial noise. We hypothesize the reason is that correcting noisy pairs in training does not help to combat adversarial noise in data.

## 4.4 LINEAR PROBING EVALUATION

We further perform evaluations on linear probing classification tasks, wherein we fit a linear classifier with a downstream training dataset by leveraging the fixed learned visual encoder. The finetuned

Table 3: Linear Probing Top1 Classification Cccuracy (%) on Vision Benchmarks.

| | Caltech101 | SVHN | STL10 | CIFAR10 | CIFAR100 | DTD | FGVCAircraft | OxfordPets | SST2 | Food101 | GTSRB | StanfordCars | Flowers102 | Average |
|---|---|---|---|---|---|---|---|---|---|---|---|---|---|---|
| CLIP | 79.3 | 45.9 | 88.7 | 76.1 | 54.1 | 55.9 | 21.4 | 57.8 | 54.2 | 55.2 | 68.2 | 78.1 | 17.7 | 57.9 |
| DeCL | 76.5 | 40.9 | 89.2 | 75.3 | 52.7 | 56.3 | 19.8 | 56.1 | 53.6 | 53.0 | 66.8 | 73.3 | 15.4 | 56.1 |
| OURS | **81.4** | **49.2** | **89.9** | **77.4** | **55.5** | **58.0** | **23.8** | **62.1** | **56.8** | **59.0** | **73.9** | **80.5** | **19.3** | **60.5** |

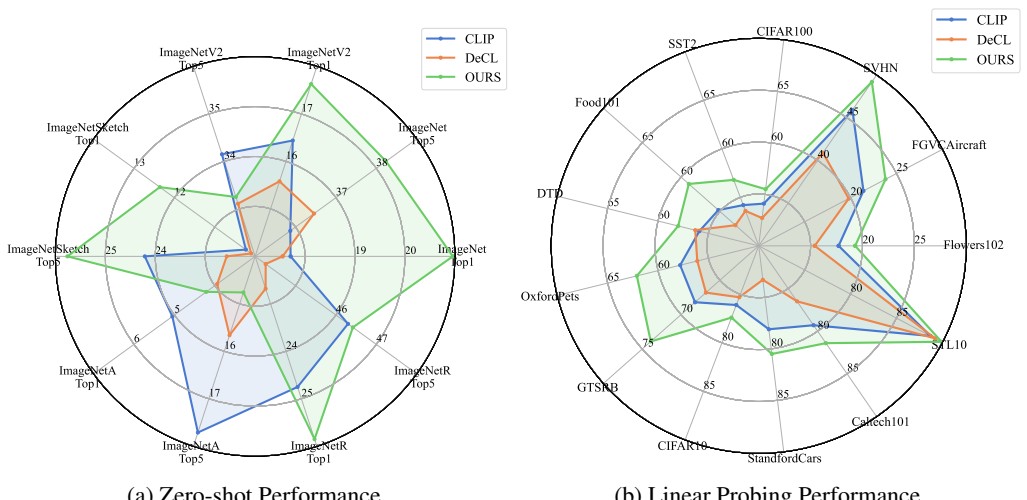

(a) Zero-shot Performance.    (b) Linear Probing Performance.

Figure 2: Visualization of model performance. Every axis denotes the performance on a particular dataset measured using wither Top1 or Top5 accuracy metric. Distinct colors signify different methods or approaches. An approach that spans a larger area demonstrates superior overall performance.

model is then evaluated on the testing dataset. This setting is used to evaluate how well the learned embeddings can generalize to new tasks with further supervision that requires only minimum fine-tuning effort. Following standard setup, we test on 14 standard benchmarks (Krizhevsky, 2009; Russakovsky et al., 2015; Fei-Fei et al., 2006; Netzer et al., 2011; Coates et al., 2011; Cimpoi et al., 2014; Maji et al., 2013; Parkhi et al., 2012; Socher et al., 2013; Bossard et al., 2014; Houben et al., 2013; Krause et al., 2013; Nilsback & Zisserman, 2008).

As shown in Table 3, our method outperforms both CLIP and DeCL on all the datasets, leading to an average gain of 3-4%. This further validates that our approach enables more flexible training with a higher tolerance for noisy data pairs, which can improve the model performance for better representations.

We visualize the model performance in Figure 2 where each color represents a different approach and the larger the area one approach covers indicates the better performance. We can see that our method outperforms baselines on both tasks with more advantage on linear probing tasks.

## 4.5  ANALYSIS

We perform analysis to further investigate our approach. We first test the sensitivity of our method on different sampling parameters. As shown in Section 2.2 and Algorithm 1, there are several hyper-parameters of the two Gamma distributions that need to be determined. Following the same setting as in DeCL we introduce a Gamma prior for $u_i$'s with the shape and rate parameters being $a_u = 1$ and $b_u = 0$. We then choose the parameters for the prior Gamma distribution for $\mathbf{w}$, where we need to determine $a_-$ and $b_-$ for the negative pairs as well as $a_+$ and $b_+$ for positive pairs. For simplicity and without loss of generality we set $b_-$ and $b_+$ to be 0. To reduce the search space, we simply fix

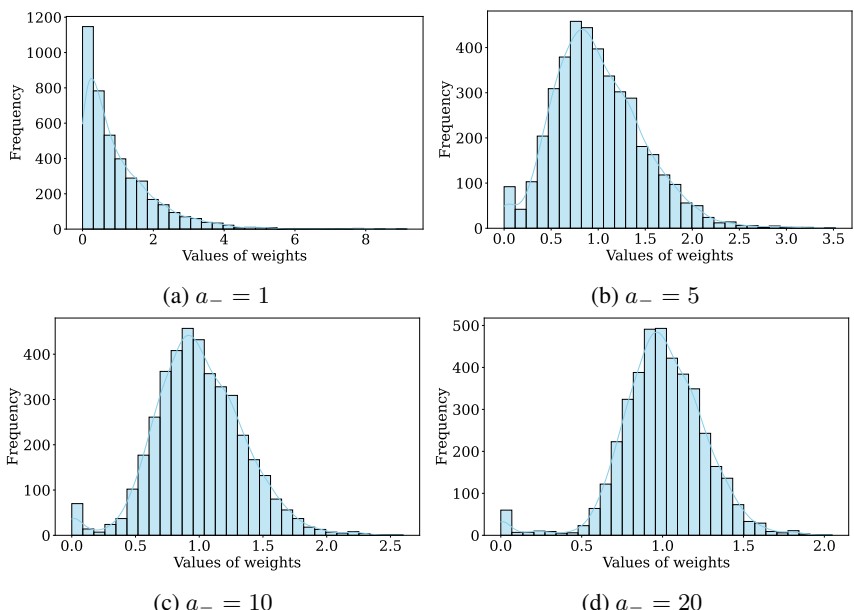

Figure 3: Posterior sample distribution of pair weights $\mathbf{w}$ with different prior choices, where $a_+ = 5$. $a_- = 10$ features the best performance.

Table 4: Effect of Changing Sampling Parameters on ImageNet zero-Shot Classification (%).

|  | $a_- = 1$ | | $a_- = 5$ | | $a_- = 10$ | | $a_- = 20$ | |
|---|---|---|---|---|---|---|---|---|
|  | Top1 | Top5 | Top1 | Top5 | Top1 | Top5 | Top1 | Top5 |
| $a_+ = 5$ | 18.00 | 34.57 | 18.02 | 34.55 | **20.96** | **38.24** | 18.39 | 35.38 |

$a_+$ and grid search for the best value of $a_-$. We set $a_+ = 5$ and search over $\{1, 5, 10, 20\}$ for $a_-$, where a higher value prefers higher weight in prior on negative pairs.

The corresponding results are shown in Table 4. As we can see, the optimal value for $a_-$ is twice of $a_+$ with the trend that neither higher or lower value brings greater gain. This indicates that slightly higher weights on negative pairs are preferable in the noisy dataset training scenarios while paying too much attention to negative pairs is not desirable as it might mitigate the learning signal from positive pairs. We also visualize the learned distribution sample results in Figure 3. We can observe that by properly setting the hyper-parameters, most of the sampled weights lie around 1 and there are pairs that are associated with much higher weights or lower weights. This observation is expected as our goal is to enable the model to have extra adaptation to automatically determine to lower weights for noisy training pairs.

## 5 CONCLUSION

In this paper, we investigate an important yet unnoticeable limitation of standard contrastive learning, where data come with noisy positive-negative pairs. Standard CL cannot handle this problem as it treats each pair equally. As a remedy, we propose a principled solution to CL by reformulating it into a probability framework and introducing random weights for data pairs. With a Bayesian data augmentation technique, the random weights can be efficiently inferred via sampling, and the model parameter can be effectively optimized via stochastic expectation maximization. The effectiveness of our innovative approach has been proven through rigorous evaluations on standard benchmarks, including applications in multi-modal contrastive learning based on the CLIP framework. The results also showcase the wide-ranging applicability and improved robustness of our proposed method. We believe our method is a valuable addition to the literature on contrastive representation learning, which can further boost the performance of state-of-the-art representation learning foundation models with larger datasets.

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
