# OpenReview forum: "Learning Multi-Modal Representation Alignments from Noisy Data-Pairs"
_ICLR.cc/2024/Conference — ICLR 2024 Conference Withdrawn Submission_

### Official Review · Reviewer_LvQG · 2023-10-21

**Soundness:** 2 fair
**Presentation:** 3 good
**Contribution:** 2 fair
**Rating:** 3
**Confidence:** 4

**Summary:**

This paper aims to solve the noise pair phenomenon that exists in multi-modal contrastive learning, so as to avoid these noise samples from affecting the training of the model.The paper proposes a re-weighting method to set different weights for different samples, and designs a weight acquisition scheme. Some experiments were conducted to verify the effectiveness of the proposed method.

**Strengths:**

1. The paper is simple and easy to understand
2. The proposed method is reasonable

**Weaknesses:**

The paper misses some important references, such as DataComp: In search of the next generation of multimodal datasets, which proposes many baseline methods to solve noisy samples in multimodal contrastive learning. For example, the simplest solution is to use CLIP after training to filter the training samples. Due to the lack of discussion and comparison of these key and closely related methods, I believe that this paper cannot be accepted.
At the same time, the most important thing is that, in my opinion, the method proposed in the paper lacks novelty. The basic idea of the proposed scheme is common in solving learning with noisy samples.

**Questions:**

None

---

> ### Author Response · Authors · 2023-11-23
> **Response from authors**
>
> **Compare to DreamData:** We thank the reviewer for the feedback. We contend that our approach diverges from the focus on data filtering, as seen in DataComp [9]. Instead, our method is designed to address the challenge of handling noisy data pairs within the dataset during training. These are distinct problems with different emphases. While we acknowledge the benefits of data filtering in terms of producing a cleaner dataset, we will incorporate discussions in the related works section, recognizing its relevance to distinct problem domains.
>
> **Novelty:** Our method is taken from the perspective of Bayesian learning, which is novel and not considered by existing works in the domain of self-supervised learning. Estimating the variance of predictions is an inherited advantage of Bayesian models. This is because in Bayesian models, the variables of interest (e.g., the weights of the data pairs) is in the form of a distribution. Given the prior and likelihood, their posterior distributions can be effectively calculated. And given the posterior distribution, its confidence (e.g., in the form of variance) can be directly estimated. We will add more background knowledge of Bayesian inference in the revision.
>
> **References:**
> Robust Cross-Modal Representation Learning with Progressive Self-Distillation use teach-student distillation for soft-alignment of data pairs.[1]
> RINCE: rank the image-text data pairs into different degrees of similarity for robust learning.[2]
> Cross-Modal Retrieval with Partially Mismatched Pairs: utilize negative pairs to tackle situation with mismatched pairs.[3]
> OpenCLIP: https://github.com/mlfoundations/open_clip [4]
> Understanding and Constructing Latent Modality Structures in Multi-Modal Representation Learning.[5]
> CyCLIP: Cyclic Contrastive Language-Image Pretraining.[6]
> Scaling Up Visual and Vision-Language Representation Learning With Noisy Text Supervision. [7]
> BLIP: Bootstrapping Language-Image Pre-training for Unified Vision-Language Understanding and Generation. [8]
> DataComp: In search of the next generation of multimodal datasets. [9]

---

### Official Review · Reviewer_UYnq · 2023-10-26

**Soundness:** 3 good
**Presentation:** 3 good
**Contribution:** 2 fair
**Rating:** 3
**Confidence:** 4

**Summary:**

This paper addresses the noisy text challenge in the image text contrastive learning problem. The authors provide an intuitive idea by learning a weight of each data pair. However, this leads to an impractical solution because of the massive amount of newly introduced variables. To remedy it, the authors follow DeCL work by reformulating it into a probability framework that can be effectively optimized via stochastic expectation maximization. The results of the designed experimental setting show improvement on the standard benchmarks.

**Strengths:**

The paper presents an intuitive idea of assigning different weights according to the noisiness of data pairs. The methodology part is mostly easy to follow with a clear motivation. Despite some concerns and questions on the experimental setting, the reported results show an improvement over the baseline CLIP and DeCL.

**Weaknesses:**

1. I recognize the new part of solving the noisy text problem by formulating it into a probabilistic framework and optimized with a similar solution proposed by DeCL (Chen et al., 2022). While this still inevitably greatly limits the contribution of this work. The technical novel part is marginal.

2. Missing related works. There are several important works in vision language learning domain that are missing, including but not limited to

	[BLIP] BLIP: Bootstrapping Language-Image Pre-training for Unified Vision-Language Understanding and Generation, icml 2022

	[ALIGN] Scaling Up Visual and Vision-Language Representation Learning With Noisy Text Supervision, icml 2021

3. In the experiment, I was not quite sure why the reported CLIP results (e.g. table 1 and 2) are far lower than the numbers reported in the original CLIP or following papers. For instance, top1 accuracy of zero-shot classification on imagenet-1k, the reported CLIP is 17.71. However, its number is 76.2 reported in ALIGN (see table 4 in ALIGN). The authors mentioned the batch size difference. This huge gap makes the experiment less convincing.

   Another noticeable difference, is the added 10% random noise. I was not sure how much this part affected the final results. However it leads to another missing parameter analysis on the different added noisy levels from 0%.

4. This paper only covers one backbone, I.e. RN-50. I would like to see other backbones such as ViT variants.

5. Typo. Page 8, table 3 caption, “cccuracy”.

   Page 8, it seems only 13 datasets are evaluated, not 14.

**Questions:**

What does the index “j” mean in Eq 1 and the last equation in page 3?

---

> ### Author Response · Authors · 2023-11-23
> **Response from authors**
>
> **Novelty and comparison to DeCL:** Thank for for the comments however we want to emphasize that our method distinguishes from DeCL. Please note that DeCL is the first work to introduce Bayesian inference into contrastive learning. However, its motivation is completely different from ours: DeCL aims to resolve the gradient bias issue whereas ours aims to resolve the noisy pair problem. From the technical perspective, although our method is built upon DeCL, the main technical novelty is to introduce the “weight” random variable, which was not considered in DeCL.
>
> **Related works:** Acknowledging the significance of ALIGN [7] and BLIP[8]  in the field, we have already cited ALIGN in our paper and will include references to the BLIP series as well. It is essential to note that these frameworks, while serving as pivotal contributions to general vision-language multi-modal learning, do not specifically address the challenge of mitigating noisy data. As foundation models, both BLIP and ALIGN have been pre-trained on extensive datasets, numbering in the billions. Consequently, a fair comparison with our approach may be challenging due to the differences in dataset scale. To provide a comprehensive overview, we will include more foundational model papers in the related works section.
>
> **Experimental results:** The accuracy of 76.2% in ALIGN [7] is achieved through pre-training on a substantial dataset of 1.8 billion image-text pairs. In contrast, our pre-training utilizes CC3M, a dataset comprising only 3 million image-text pairs. It is noteworthy that our results align with the state-of-the-art (SOTA) in image-text pre-training methods when using CC3M for pre-training. Specifically, our performance is consistent with reported accuracies in the OpenAI CLIP repository [4], as well as results of 16.72% in [5] and 20.03% in [6].
>
> **Add more backbone:** We appreciate the suggestion. It is acknowledged that, owing to the substantial computational demands, many state-of-the-art (SOTA) papers often opt for a single vision encoder, such as ResNet, for pre-training. Nevertheless, we will heed the recommendation to incorporate experiments with ViT backbone in the final version.
>
> **Linear evaluation datasets:** We used ImageNet for zero-shot evaluation thus we did not perform linear probing on the same dataset. We will add the linear probing results on ImageNet into our final version.
>
> **Clarification on notation:** In Eq.1, we use index i and j to refer data within the minibatch, i.e., i refer to the i-th data in the batch, and $s_{i,j}$ represent the similarity score between the i-th image and the j-th text. We will add more details in the method section to make sure the notations and derivations are more clear and smooth.
>
> **References:**
> Robust Cross-Modal Representation Learning with Progressive Self-Distillation use teach-student distillation for soft-alignment of data pairs.[1]
> RINCE: rank the image-text data pairs into different degrees of similarity for robust learning.[2]
> Cross-Modal Retrieval with Partially Mismatched Pairs: utilize negative pairs to tackle situation with mismatched pairs.[3]
> OpenCLIP: https://github.com/mlfoundations/open_clip [4]
> Understanding and Constructing Latent Modality Structures in Multi-Modal Representation Learning.[5]
> CyCLIP: Cyclic Contrastive Language-Image Pretraining.[6]
> Scaling Up Visual and Vision-Language Representation Learning With Noisy Text Supervision. [7]
> BLIP: Bootstrapping Language-Image Pre-training for Unified Vision-Language Understanding and Generation. [8]

---

### Official Review · Reviewer_6kBB · 2023-10-31

**Soundness:** 3 good
**Presentation:** 3 good
**Contribution:** 3 good
**Rating:** 5
**Confidence:** 5

**Summary:**

This paper provides a comprehensive exploration of noise-robust contrastive learning and introduces an innovative approach for addressing noisy data pairs. The authors propose the incorporation of learnable stochastic weights, informed by a Bayesian inference framework, to dynamically adjust the significance of each data pair. To facilitate the learning of these weights, the paper reformulates the problem within a probabilistic framework and devises a stochastic expectation maximization algorithm.The paper evaluates the method on several multi-modal contrastive learning benchmarks.

**Strengths:**

This paper introduce a new method to combat with noisy data pairs through learnable stochastic weight from the Bayesian inference framework.

Extensive experiments have been conducted to verify the effectivenss of the proposed method.

**Weaknesses:**

1. One primary concern revolves around the claimed novelty of the research problem. While the authors assert that they are the first to address the issue of contrastive learning with noisy data pairs, existing studies have already explored similar territories. For instance, RINCE (CVPR 2022) focuses on mitigating the impact of noisy views (false positives) and presents a theoretically-grounded robust infoNCE loss function. More recent studies in noisy correspondence learning tackle issues like mismatched pairs and false negatives in diverse tasks, including partially view-aligned clustering, video-text retrieval, visible-infrared person re-identification, and image-text matching. Furthermore, [1] formally investigates noisy many-to-many correspondences and introduces an innovative training framework that outperforms its CLIP counterpart in multiple settings.I believe it is crucial to provide a more in-depth clarification of the distinctions between this paper and previous works. Additionally, it would be valuable to expand the related works section to offer a deeper understanding of these differences.
   [1] Robust Cross-Modal Representation Learning with Progressive Self-Distillation, CVPR 2022

2. The paper lacks sufficient detail and background information. Notably, explanations and prerequisites related to concepts such as the gamma distribution/prior and Bayesian inference are notably absent. Furthermore, some equations are inadequately elucidated and appear to be missing essential derivations.
3. The paper's comparative analysis is limited to CLIP and DeCL, neglecting other crucial baselines, like [1], which specifically address the same issue of noisy data pairs.
4. It is recommended to supplement the paper with additional experimental analyses that illuminate the inner workings of the proposed method. For instance, incorporating visualizations of the learned weights for positive and negative pairs, akin to Fig. 5 in [1], would enhance the reader's understanding of the method's underlying mechanisms.

**Questions:**

Please see the weaknesses

---

> ### Author Response · Authors · 2023-11-23
> **Response from authors**
>
> **Related works:** Thank you for the valuable comments and suggestions for related works. We acknowledge the significance of incorporating discussions on these works in both the related works section and the experimental comparison. Specifically, in the case of RINCE [2], it achieves robustness by ranking data pairs into different degrees of positives. However, it is essential to note that RINCE necessitates additional information/ground truth, such as super-classes and sub-classes in the CIFAR dataset, for ranking positives. This requirement may not always be practical in the context of large-scale vision-language datasets. In contrast, our approach, rooted in Bayesian inference, does not rely on any supplementary information to reweight data pairs. While RINCE addresses the same issue of contrastive learning concerning noisy one-to-one mapping data pairs, our method distinguishes itself by its independence from additional information requirements. We will incorporate RINCE as a baseline for comparison in the final version of our work.
>
> **Background information:** Our model is taken from the perspective of Bayesian learning, which is novel and not considered by existing works in the domain of self-supervised learning. In Bayesian inference, we assign a prior distribution for random variables of interest (the Gamma distribution for w in our case), which stands as our prior belief on the value of the random variables. Together with the likelihood from the loss function, we can approximate the posterior distribution. Bayesian inference is the method that aims to approximate the posterior distribution of the random variables efficiently (in our case, we resort the sampling). Thanks for the comment, we will add sufficient background into our revision and make the paper more clear.
>
> **Visualization:** Thank you for the recommendation. In Figure.3 we visualize the weight distribution of our approach . We will indeed incorporate your suggestion by extending the visualization to include similarity scores from different approaches. We will provide a more comprehensive illustration of the effectiveness of our method.
>
> **References:**
> Robust Cross-Modal Representation Learning with Progressive Self-Distillation use teach-student distillation for soft-alignment of data pairs.[1]
> RINCE: rank the image-text data pairs into different degrees of similarity for robust learning.[2]
> Cross-Modal Retrieval with Partially Mismatched Pairs: utilize negative pairs to tackle situation with mismatched pairs.[3]
> OpenCLIP: https://github.com/mlfoundations/open_clip [4]
> Understanding and Constructing Latent Modality Structures in Multi-Modal Representation Learning.[5]
> CyCLIP: Cyclic Contrastive Language-Image Pretraining.[6]
> Scaling Up Visual and Vision-Language Representation Learning With Noisy Text Supervision. [7]
> BLIP: Bootstrapping Language-Image Pre-training for Unified Vision-Language Understanding and Generation. [8]

---

> > ### Comment · Reviewer_6kBB · 2023-11-23
> >
> > I appreciate your response. However, based on my understanding, RINCE doesn't require information or ground truth. Additionally, my concern regarding the differences between the referenced related works and this study remains unaddressed. Therefore, I maintain my negative assessment.

---

### Official Review · Reviewer_g84h · 2023-11-01

**Soundness:** 2 fair
**Presentation:** 2 fair
**Contribution:** 2 fair
**Rating:** 3
**Confidence:** 5

**Summary:**

This paper addresses the issue of noisy pairs in vision-language pre-training by proposing a weighted contrastive loss that estimates the noisiness of data pairs using Bayesian techniques. The proposed method is evaluated on the CLIP model and demonstrates reasonable improvements compared to vanilla CLIP.

**Strengths:**

1. The problem of addressing noisy pairs in vision-language pre-training has practical implications and is valuable.
2. The proposed method takes into account both the false positive and false negative problems in contrastive learning.

**Weaknesses:**

1. There is a concern regarding the originality of the problem studied in this paper. The claim of being the first to consider the noise problem in contrastive learning is incorrect, as several prior works have already addressed this issue.: Robust Cross-Modal Representation Learning with Progressive Self-Distillation, Ranking info noise contrastive estimation: Boosting contrastive learning via ranked positives, Cross-Modal Retrieval with Partially Mismatched Pairs and Debiased Contrastive Learning, to name a few.

2. Given the focus on noisy pairs, the related works section should provide a comprehensive discussion covering noisy labels, noisy correspondence, and robust vision-language pre-training. Moreover, the approach of incorporating soft weights into contrastive learning is not novel in either the noisy label or vision-language learning community.

3. The proposed method does not show notable improvements compared to vanilla CLIP, particularly according to Table 2.

4. The estimation of the confidence of noisy pairs through the joint posterior distribution over the global model parameter and local random weight (Eq. 2) is still unclear. It lacks sufficient details and intuitive explanation, making it difficult to understand how the confidence of noisy pairs is estimated.

**Questions:**

In the implementation details, the authors manually simulate around 10% noisy pairs. The rationale behind this choice is unclear. It would be more informative to evaluate the method on naturally occurring partially-matched pairs, such as the 3% to 22% range found in CC. Additionally, exploring performance on even noisier datasets with 30% or 50% noisy pairs would provide a more challenging scenario for evaluation if the simulate noisy pair is needed.

---

> ### Author Response · Authors · 2023-11-23
> **Response from authors**
>
> **Related works:** Thank you for the valuable comments and suggestions for related works. We will add the discussion of these works into related works as well as comparison in the experimental section. [1] applies teacher-student network in a self-distillation fashion to generate soft targets for a partition of data pairs in the batch, so that many-to-many mapping can be achieved. Nevertheless, it is crucial to highlight that this method remains dependent on the current model for the generation of soft targets. In contrast, our stochastic approach, rooted in Bayesian inference, is inherently independent of the model.  RINCE [2] achieves robustness through ranking the data pairs into different degree of positive; however it requires additional information/groundtruth to rank positives such as super-classes and sub-classes in CIFAR dataset, which is not always feasible in large-scale vision-language datasets. Our approach does not require such additional information to achieve robust models. [3] leverages negative samples as complementary learning to alleviate the impact of partially mismatched pairs. In contrast, our approach involves the reweighting of all data pairs. These baseline methods will be included for comprehensive comparison in the final version of our work.
>
> **Experimental results:** In Table 2, our method is assessed against distribution shift benchmarks, demonstrating superior performance on three out of the four datasets compared to the state-of-the-art (SOTA). It is important to highlight that across various benchmarks, such as zero-shot evaluations (Table 1) and linear probing assessments (Table 3), our approach consistently outperforms the current state-of-the-art by a significant margin. It is noteworthy that, among the diverse tasks and 18 evaluation datasets encompassed, our method exhibits only a marginal decrease in performance on one dataset within the distribution shift benchmark—an outcome we deem reasonable.
>
> **Novelty of approach:** Our model is taken from the perspective of Bayesian learning, which is novel and not considered by existing works in the domain of self-supervised learning. In Bayesian inference, we assign a prior distribution for random variables of interest (the Gamma distribution for w in our case), which stands as our prior belief on the value of the random variables. Together with the likelihood from the loss function, we can approximate the posterior distribution. Bayesian inference is the method that aims to approximate the posterior distribution of the random variables efficiently (in our case, we resort the sampling). Estimating the variance of predictions is an inherited advantage of Bayesian models. This is because in Bayesian models, the variables of interest (e.g., the weights of the data pairs) is in the form of a distribution. Given the prior and likelihood, their posterior distributions can be effectively calculated. And given the posterior distribution, its confidence (e.g., in the form of variance) can be directly estimated. We will add more background knowledge of Bayesian inference in the revision.
>
> **Increase noise level:** Thank you for the suggestion. Indeed we performed experiments with increased noise levels to test the performance. We will include the following results in the supplemental materials:
>
> | Method | 10% Noise| - | 30% Noise | - | 50% Noise | - |
> | ----------- | ----------- | ----------- | ----------- | ----------- | ----------- | ----------- |
> |  | Top-1 | Top-5 | Top-1 | Top-5 | Top-1 | Top-5 |
> | CLIP | 17.71 | 35.87 | 20.06 | 38.14 | 18.05 | 38.13 |
> | OURS | **20.96** | **38.24** | **20.96** | **38.24** | **19.91** | **38.49** |
>
> **References:**
> Robust Cross-Modal Representation Learning with Progressive Self-Distillation use teach-student distillation for soft-alignment of data pairs.[1]
> RINCE: rank the image-text data pairs into different degrees of similarity for robust learning.[2]
> Cross-Modal Retrieval with Partially Mismatched Pairs: utilize negative pairs to tackle situation with mismatched pairs.[3]
> OpenCLIP: https://github.com/mlfoundations/open_clip [4]
> Understanding and Constructing Latent Modality Structures in Multi-Modal Representation Learning.[5]
> CyCLIP: Cyclic Contrastive Language-Image Pretraining.[6]
> Scaling Up Visual and Vision-Language Representation Learning With Noisy Text Supervision. [7]
> BLIP: Bootstrapping Language-Image Pre-training for Unified Vision-Language Understanding and Generation. [8]